# Learning Distributed Geometric Koopman Operator for Sparse Networked Dynamical Systems

**Sayak Mukherjee**[1], **Sai Pushpak Nandanoori**[1], **Sheng Guan**[2], **Khushbu Agarwal**[1], **Subhrajit Sinha**[1], **Soumya Kundu**[1], **Seemita Pal**[1], **Yinghui Wu**[2], **Draguna L Vrabie**[1], **Sutanay Choudhury**[1]

[1] Pacific Northwest National Laboratory, Richland, Washington, USA
[2] Case Western Reserve University, Cleveland, Ohio, USA
{ sayak.mukherjee, saipushpak.n, khushbu.agarwal, subhrajit.sinha, soumya.kundu,
draguna.vrabie, sutanay.choudhury}@pnnl.gov
{sxg967, yxw1650 }@case.edu

## Abstract

The Koopman operator theory provides an alternative to studying nonlinear networked dynamical systems (NDS) by mapping the state space to an abstract higher dimensional space where the system evolution is linear. The recent works show the application of graph neural networks (GNNs) to learn state to object-centric embedding and achieve centralized block-wise computation of the Koopman operator (KO) under additional assumptions on the underlying node properties and constraints on the KO structure. However, the computational complexity of learning the Koopman operator increases for large NDS. Moreover, the computational complexity increases in a combinatorial fashion with the increase in number of nodes. The learning challenge is further amplified for sparse networks by two factors: 1) sample sparsity for learning the Koopman operator in the non-linear space, and 2) the dissimilarity in the dynamics of individual nodes or from one subgraph to another. Our work aims to address these challenges by formulating the representation learning of NDS into a multi-agent paradigm and learning the Koopman operator in a distributive manner. Our theoretical results show that the proposed distributed computation of the geometric Koopman operator is beneficial for sparse NDS, whereas for the fully connected systems this approach coincides with the centralized one. The empirical study on a rope system, a network of oscillators, and a power grid show comparable and superior performance along with computational benefits with the state-of-the-art methods.

## 1 Introduction

NDS represents an important class of dynamic networks where the state of the network is defined by a vector of node-level properties in a geometrical manifold, and their evolution is governed by a set of differential equations. Data-driven modeling of both spatio-temporal dependencies and evolution dynamics is essential to predict the response of the NDS to an external perturbation. Surely, machine-learning approaches that explicitly recognize the interconnection structure of such systems or model the dynamical system-driven evolution of the network outperform initial deep learning approaches based on recurrent neural networks and its variants [1–5]. Deep learning approaches such as GNNs fit into this paradigm by learning non-linear functions for each of the encoder-system model-decoder components [6–10]. Discovering the underlying physics of dynamical systems have intrigued control theory researcher for decades resulting into multiple sub-space based system identification works [11–14]. Koopman operator theory [15, 16] is an approach for such model discovery where the core idea is to transform the observed state-space variables to the space of square-integrable functions, where a linear operator provides an exact representation of the underlying dynamical system and the spectrum of the operator encodes all the non-linear behaviors. However, for computational purposes,

S. Mukherjee et al., Learning Distributed Geometric Koopman Operator for Sparse Networked Dynamical Systems. *Proceedings of the First Learning on Graphs Conference (LoG 2022)*, PMLR 198, Virtual Event, December 9–12, 2022.

finding finite-dimensional approximation of Koopman operator is challenging. The key to computing the finite-dimensional Koopman operator is fixing the lifting functions (observables) and existing approaches such as classical or extended dynamic mode decomposition [17, 18] use an a-priori choice of basis functions for lifting; however, this choice usually fails to generalize to more complex environments. Instead, learning these transformations from the system trajectories themselves using deep neural networks (DNNs) have been shown to yield much richer invariant subspaces [19, 20].

Continuing the idea of lifting the non-linear state space into another space to learn linear transition dynamics, [21] proposed the use of a graph neural network as the encoder-decoder function. While graph neural networks (GNN) [22] appears to be a natural approach for modeling the physics of networked systems, their ability to discover dynamic evolution models of large-scale networked systems is a nascent area of research [6, 7, 9, 23]. For NDS, where the number of system states increases with the number of nodes, the computational complexity of learning the Koopman operator also increases. The topology of the network or its sparsity are typically not taken advantage of in the existing studies when learning observable functions or the Koopman operator.

In this work, we address the challenge of learning dissimilar dynamics in sparse networks by formulating the representation learning of networked dynamical systems into a multi-agent paradigm. We refer to this approach as Distributed Koopman-GNN (DKGNN). DKGNN is more suitable for sparse and large networked dynamical systems as the proposed distributed learning method yields superior computational efficiency compared to traditional methods. We applied the GNNs to capture the distributed nature of the dynamical system behavior, transform the original state-space into the Koopman observable space, and subsequently use the network sparsity patterns to constrain the Koopman operator construction into a block-structured distributed representation along with theoretical guarantees. Information-theoretic network clustering strategies were utilized for specific dynamic systems to capture the joint evolution of the clusters in a coarse-grained fashion resulting in further computational benefits. Please see Figure 1 for an illustration of the approach.

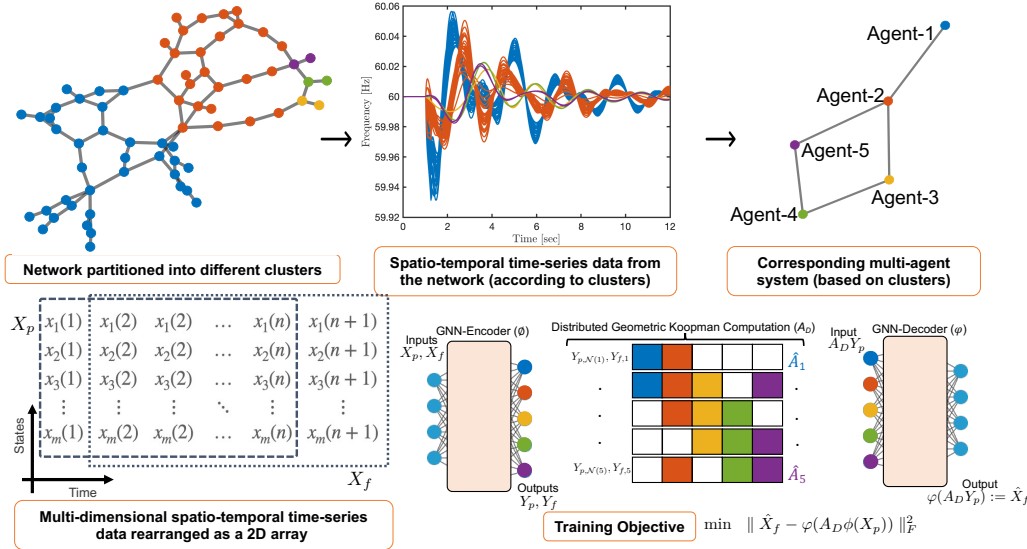

**Figure 1:** Overview of the proposed approach (best viewed with colors) : (Top row) the sparse networked dynamical system (NDS) is partitioned into clusters using dynamic spatio-temporal data resulting into an agent representation (each color represents an agent). The time-series associated with each node is also color coded by the agents. (Bottom row) the re-arranged multi-dimensional spatio-temporal data is fed to the GNN along with the agent network structure to learn the nonlinear observables. Learning the distributed geometric Koopman operator by exploiting the sparsity of the multi agent system is shown in lower right, with colors in the Koopman matrix capturing the distributed connection in the agent topology (white blocks correspond to no edges between the agents and hence they are all zeros).

In this work, we compared with [21] which uses object-centric embeddings and learns a centralized KO without considering the network structure. We use advanced clustering methods that consider the

dynamical evolution and exploit the resultant network structure to obtain a distributed KO. Moreover, [21] deal with relatively small-sized systems, whereas we have extended it to large-scale NDS. [20] uses a feed forward NN (MLP) to learn the observables that are state-inclusive and hence, don't need to train a decoder separately, whereas, we use GNN-encoder and simultaneously train a GNN-decoder. In [19], the authors also use generic neural network-based autoencoders for learning the Koopman observables. Moreover, they assume that the eigenfunctions are the Koopman observables which is difficult to ensure for pure data-driven learning of dynamical systems, and in this work, we propose a more general framework without this conservative assumption.

**Contributions.** The main contributions of this paper are summarized as follows:

- We develop methods for learning distributed Koopman operator for large-scale networks, using system topology and network sparsity properties for NDS. We present a system theoretic learning approach that can exploit the network connectivity structure via GNNs.

- We introduce information theoretic-based clustering strategies for sparse NDS to learn a coarsened structure and model the system using a hierarchical multi-agent paradigm.

- We present theoretical results on bounding the performance of the distributed geometric Koopman operator with respect to its centralized counterpart.

- We demonstrate that DKGNN yields two benefits. It improves the scalability of learning, and for sparse NDS with dissimilar dynamics across different parts of the network, it outperforms prediction performance of centralized approaches.

## 1.1 Related Work

**Koopman operator theory** The infinite-dimensional Koopman operator is computationally intractable. Several methods for identifying approximations of the infinite-dimensional Koopman operator on a finite-dimensional space have recently been developed. Most notable works include dynamic mode decomposition (DMD) [17, 24], extended DMD (EDMD) [18, 25], Hankel DMD [26], naturally structured DMD (NS-DMD) [27] and deep learning based DMD (deepDMD) [20, 28]. These methods are data-driven and one or more of these methods have been successfully applied for system identification [17, 29] including system identification from noisy data [30], data-driven observability/controllability gramians for nonlinear systems [31, 32], control design [33–35], data-driven causal inference in dynamical systems [36] and to identify persistence of excitation conditions for nonlinear systems [37]. [38] discusses distributed design of Koopman without control and using dictionary lifting functions.

**Graph Neural Networks** GNNs [22, 39] have found widespread use into every application involving non-Euclidean data [40]. Extending GNNs to model physics-driven processes gives rise to a new class of physics-inspired neural networks (PINN) [8–10]. A common theme is to model many-body interactions via a nearest-neighbor graph and then model the evolution of that graph [6, 7, 9]. However, addressing issues around compositionality [7, 23] and scalability becomes important as the foundation for PI(G)NN matures and we seek to model larger, multi-scale spatio-temporal interactions. Moreover, applications such as molecular biology [20] and power grid [41] motivate the modeling of NDS where the graph structure is distinct from k-nearest neighbor graphs, with sparsity and connectivity that resemble small-world networks. Recent works such as [21] provides a bridge that seeks to integrate GNNs and Koopman operators to improve generalization ability and result in simpler linear transition dynamics. However, their approach for learning Koopman state transitions and GNN embedding results in performance and scalability bottlenecks when system size increases.

## 2 Methodology

### 2.1 Networked Dynamical Systems and the Koopman Operator

**Problem Statement**: Consider a networked dynamical system (NDS) evolving over a network, $\mathcal{G} = (\mathcal{V}, \mathcal{E})$. Let the number of nodes and edges be $n_v$ and $n_e$ respectively and the governing equation for the NDS on $\mathcal{G}$ is given by,

$$x_{t+1} = F(x_t), \tag{1}$$

where $x_t \in \mathcal{M} \subseteq \mathbb{R}^n$ is the concatenated system state at time $t$ and $F : \mathcal{M} \to \mathcal{M}$ is the discrete-time nonlinear transition mapping. Our goal is to learn the system dynamics as expressed in equation (1)

in a distributed approach combining the Koopman operator theory, graph neural networks and by leveraging on network sparsity properties.

Exploring the network structure, the $n_v$ nodes could be grouped to form $n_a$ agents where $n_a \leq n_v$ and results in a network denoted by $\mathcal{G}_a = (\mathcal{V}_a, \mathcal{E}_a)$ with the state at any time $t$ is partitioned as $x_t = [x_{t,1}^\top, \ldots, x_{t,n_a}^\top]^\top$ where for every $\alpha \in \{1, \ldots, n_a\}$, the states $x_{t,\alpha}$ belongs to agent $\alpha$. For completion, we mention that the number of nodes in $\mathcal{V}_a$ is equal to $n_a$. The motivation behind exploring the network structure is to develop models that will possess certain advantages when compared to the centrally learned models. A method to identify $\mathcal{G}_a$ from $\mathcal{G}$ for practical dynamical systems is discussed later in the paper. Associated with the system (1) is a linear operator, namely the Koopman operator $\mathbb{U}$ [42] which is defined as follows.

**Definition 1** (Koopman Operator (KO) [42]). *Given any $h \in L^2(\mathcal{M})$, the Koopman operator $\mathbb{U} : L^2(\mathcal{M}) \to L^2(\mathcal{M})$ for the system (1) is defined as $[\mathbb{U}h](x) = h(F(x))$, where $L^2(\mathcal{M})$ is the space of square integrable functions on $\mathcal{M}$.*

Originally developed for autonomous systems, recently Koopman framework has been extended to systems with control [43, 44]. In this paper we consider a controlled dynamical system of the form:

$$x_{t+1} = F(x_t) + G(x_t)u_t, \tag{2}$$

where $G : \mathcal{M} \to \mathbb{R}^{n \times q}$ is the input vector field and $u_t \in \mathbb{R}^q$ denote the control input to the system at time $t$. The Koopman operator associated with (2) is defined on an extended state-space obtained as the product of the original state-space and the space of all control sequences, resulting in a control-affine dynamical system on the extended state-space [43, 44]. In general, the Koopman operator is an infinite-dimensional operator, but for computation purposes, a finite-dimensional approximation of the operator is constructed from the obtained time-series data as discussed below.

Consider the time-series data from a networked dynamical system as $X = [x_1 \quad x_2 \quad \ldots \quad x_k] \in \mathbb{R}^{n \times k}$, and the corresponding control inputs $U = [u_1 \quad u_2 \quad \ldots \quad u_k] \in \mathbb{R}^{q \times k}$. Define one time-step separated datasets, $X_p$ and $X_f$ from $X$ as $X_p = [x_1, x_2, \ldots, x_{k-1}]$, $X_f = [x_2, x_3, \ldots, x_k]$ and let $\mathcal{S} = \{\Psi_1, \ldots, \Psi_m\}$ be the choice of non-linear functions or observables where $\Psi_i \in L^2(\mathbb{R}^n, \mathcal{B}, \mu)$ (where $\mathcal{B}$ is the Borel $\sigma$ algebra and $\mu$ denote the measure [42]) and $\Psi_i : \mathbb{R}^n \to \mathbb{C}$. Define a vector valued observable function $\Psi : \mathbb{R}^n \to \mathbb{C}^m$ as, $\Psi(x) := [\Psi_1(x) \quad \Psi_2(x) \quad \cdots \quad \Psi_m(x)]^\top$. Then the following optimization problem which minimizes the least-squares cost yields the Koopman operator and the input matrix.

$$\min_{A,B} \parallel Y_f - AY_p - BU \parallel_F^2 \tag{3}$$

where $Y_p = \Psi(X_p) = [\Psi(x_1), \cdots, \Psi(x_{k-1})]$, $Y_f = \Psi(X_f) = [\Psi(x_2), \cdots, \Psi(x_k)]$, $A \in \mathbb{R}^{m \times m}$ is the finite dimensional approximation of the Koopman operator defined on the space of observables and the matrix $B \in \mathbb{R}^{m \times q}$ is the input matrix. The optimization problem (3) can be solved analytically and the approximate Koopman operator and the input matrix are given by $[A \quad B] = Y_f[Y_p \quad U]^\dagger$ [43], where $(\cdot)^\dagger$ is the Moore-Penrose pseudo-inverse of a matrix. Identifying the observable functions such that $\mathcal{S}$ is invariant under the action of the Koopman operator is challenging. In this work, graph neural network-based mappings are used to construct the non-linear observable functions that satisfy the invariance by simultaneously learning the observables and the Koopman operator.

## 2.2 Graph Neural Network based Koopman Observables

Consider the network $\mathcal{G}$ with $n_v$ nodes where the time-series data at each node is supplemented with the node attribute capturing the nature of the node, denoted by the vector $x_{v_i}$ where $i = \{1, 2, \ldots, n_v\}$. For instance, we can characterize the generators in a electric power grid network with their inertia values. Similarly, the designer can embed knowledge about the interaction between the agents using edge attributes, denoted as $x_{e_{ij}}$ for the edge connecting nodes $i$ and $j$. We consider a graph neural network embedding to transition from the actual state-space to the lifted state-space using multiple compositional neural operations. At the $t^{th}$ time-step, the node, and edge attributes are combined along with the state vectors of the agents which are compactly written as,

$$x_{t,i}^k = f_v^k(x_{t,i}^{k-1}, \sum_{j \in \mathcal{N}(i)} f_e^k(x_{t,i}^{k-1}, x_{v_i}^{k-1}, x_{t,j}^{k-1}, x_{v_j}^{k-1}, x_{e_{ij}}^{k-1})) \tag{4}$$

where the superscript $k$ denotes the $k^{th}$ layer of the GNN, and functions $f_e(\cdot)$, and $f_v(\cdot)$ are edge and node-level aggregation functions in a GNN architecture. We use $\phi(\cdot)$ to denote the multi-layer GNN operation in a compact form. The GNN can represent a complex nonlinear mapping that is not restricted by any limitation on the message-passing layers and can capture the influence coming from further nodes of the NDS. We do not enforce any limitation on the influence propagation for GNN layers. In summary, performing the lifting from the actual state space to the Koopman state space, we are using the GNN mappings in such a way that the identity of the nodes is still maintained, and thereafter we are using the agent-level information to group them to perform the distributed Koopman design.

## 3 Distributed Geometric Koopman Operator with Control Inputs

This section formally presents the computation of distributed geometric Koopman operator with control. The (centralized) Koopman operator with control input for the system (2) is obtained by solving (3). For the $n_a$ agent NDS, the resultant KO can be represented as $n_a^2$ block matrices:

$$A = \begin{bmatrix} A_1 \\ \hline A_2 \\ \hline \vdots \\ \hline A_{n_a} \end{bmatrix} = \begin{bmatrix} A_{11} & A_{12} & \cdots & A_{1n_a} \\ \hline A_{21} & A_{22} & \cdots & A_{2n_a} \\ \hline \vdots & \vdots & \ddots & \vdots \\ \hline A_{n_a 1} & A_{n_a 2} & \cdots & A_{n_a n_a} \end{bmatrix} \tag{5}$$

The dynamics of the $\alpha^{th}$ agent have the dimension, $m_\alpha$, such that, $\sum_{\alpha=1}^{n_a} m_\alpha = m$. It now follows that the block matrix $A_{\alpha\beta} \in \mathbb{R}^{m_\alpha \times m_\beta}$ denotes the transition of agent $\alpha$ with respect to $\beta$ and the transition mapping for agent $\alpha$ is given by $A_\alpha$. Similarly, the control input matrix is partitioned as $B = blkdiag(B_1, B_2, \ldots, B_{n_a})$, where the matrix $B_\alpha$ corresponds to input matrix of agent $\alpha \in \{1, 2, \ldots, n_a\}$. The objective of the distributed learning is to compute these block matrices in a distributed manner and form the geometric Koopman operator and the control input matrix for the complete NDS as opposed to directly solving the centralized optimization problem in Eq. (3) without sacrificing the performance with the distributed method. There are two major advantages to this approach. Firstly, if there is change in the local agent behavior, one can simply update the transition mapping corresponding to that agent and the agents dependent on it to learn the full system evolution. Secondly, computational advantages can be obtained by incorporating parallel learning of each agent transition mapping and this approach is more appropriate for the sparse networks. Please note our focus is to was to exploit the dynamical system properties to exploit distributed design recognizing the advances in the scalable matrix computations on any multi-core, GPU or distributed computing framework (such as Apache Spark) today [45–48].

By exploiting the topology of the network, the KO and the control input matrices are computed in a distributed manner. As a consequence, if agent $i$ is not a neighbor of agent $j$, that is, the dynamics of agent $i$ is not affected by the dynamics of agent $j$, we make $A_{ij} = 0$. Therefore, for every $\alpha \in \{1, 2, \cdots, n_a\}$, let $\hat{A}_\alpha$ be the transition mapping corresponding to the agent $\alpha$, then the distributed Koopman is given by $A_D = \begin{bmatrix} \hat{A}_1^\top & \hat{A}_2^\top & \cdots & \hat{A}_{n_a}^\top \end{bmatrix}^\top$. For a sparse network, the distributed Koopman will be a sparse matrix irrespective of the centralized Koopman being either sparse or full (Figure 1). Consider $X_p$ and $X_f$ be the one time-step separated time-series data on the state space, $\phi$ be the GNN-embedding that maps the state space data into an embedded space. Then the time-series data on the embedded space for every agent can be expressed in terms of the neighbor and non-neighbor agents.

**Remark 2.** *The one time-step forwarded time-series data corresponding to agent $\alpha$ is given by*

$A_\alpha Y_p = \begin{bmatrix} A_{\mathcal{N}(\alpha)} & A_{\overline{\mathcal{N}(\alpha)}} \end{bmatrix} \begin{bmatrix} Y_{p,\mathcal{N}(\alpha)} \\ Y_{p,\overline{\mathcal{N}(\alpha)}} \end{bmatrix}$, *where $\mathcal{N}(\alpha)$ is the set of agents containing the neighbors of*

*agent $\alpha$ and itself, $\overline{\mathcal{N}(\alpha)}$ is the set of agents who are non-neighbors of agent $\alpha$ and the (rectangular) matrices, $A_{\mathcal{N}(\alpha)}$ and $A_{\overline{\mathcal{N}(\alpha)}}$ are the transition mappings associated with the agent $\alpha$.*

Let $R_{p,\alpha}$, $R_{f,\alpha}$, and $R_{u,\alpha}$ be the transformation matrices defined in such a way that they remove zero rows of any matrix, $D$ when pre-multiplied to the matrix, $D$. Suppose if the matrix $D$ has no zero rows then the transformation matrices are identity.

**Proposition 3.** *The centralized Koopman $(A, B)$ learning problem described in Eq. (3) can be expressed as a distributed Koopman $(A_D, B_D)$ learning problem such that there exists matrices, $\hat{A}_1, \hat{A}_2, \ldots, \hat{A}_{n_a}$, $\hat{B}_1, \hat{B}_2, \ldots, \hat{B}_{n_a}$ and the distributed Koopman operator is given by*

$A_D = \begin{bmatrix} \hat{A}_1^\top & \hat{A}_2^\top & \cdots & \hat{A}_{n_a}^\top \end{bmatrix}^\top$, *input matrix is* $B_D = blkdiag(\hat{B}_1, \hat{B}_2, \ldots, \hat{B}_{n_a})$ *where for* $\alpha \in \{1, 2, \ldots, n_a\}$, $\hat{A}_\alpha = A_{\mathcal{N}(\alpha)} R_{p,\alpha}$, $\hat{B}_\alpha = B_\alpha R_{u,\alpha}$ *and* $A_{\mathcal{N}(\alpha)}, B_\alpha$ *are obtained as a solution to the optimization problem* $\min_{A_{\mathcal{N}(\alpha)}, B_\alpha} \| Y_{f,\alpha} - A_{\mathcal{N}(\alpha)} Y_{p,\mathcal{N}(\alpha)} - B_\alpha U_\alpha \|_F^2$.

From Proposition 3, with $g_t = \phi(x_t)$, $\phi$ being the GNN encoder, the distributed geometric Koopman operator system with control input is given by $g_{t+1} = A_D g_t + B_D u_t$.

**Corollary 4.** *The distributed learning problem and the centralized learning problem yield the same Koopman operator for a fully connected network.*

The proofs for Proposition 3 and Corollary 4 are included in the appendix.

### 3.1 Training Distributed Geometric Koopman Model

The state space data is mapped to the GNN-embedded space using the GNN encoder $\phi$. To retrieve the actual state space data from the GNN-embedded space, we use a decoding GNN operator such that, $\hat{x}_t = \varphi(g_t)$. The decoder $\varphi(\cdot)$ follows similar GNN architecture as encoder however it maps from the

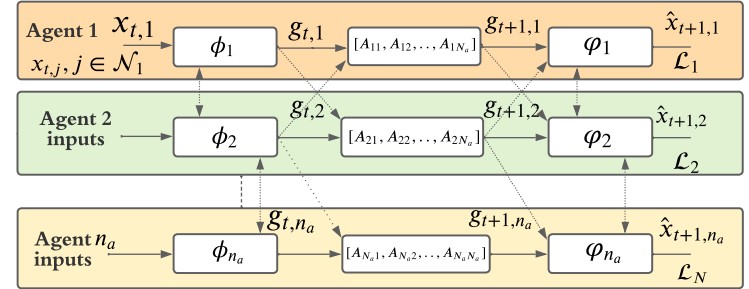

**Figure 2:** Distributed geometric Koopman architecture

lifted Koopman space to the original state space. Looking into the agent-wise architectural detail, both encoder and decoder functions can be represented for the $i^{th}$ agent as $\phi_i(\cdot), \varphi_i(\cdot)$ as shown in Figure 2 with the understanding that all the GNN functionalities take the neighbouring agent states and attributes as additional inputs. This facilitates the computation of Koopman matrices in a distributed manner. This architecture leads us to compute an auto-encoding loss, and a prediction loss over the time-steps $t = 1, 2, \ldots, k - 1$, and are given as follows:

$$\mathcal{L}_{ae} = \frac{1}{k} \sum_{t=1}^{k-1} \sum_{i=1}^{n_a} \varphi_i(\phi_i(x_{t,i})) - x_{t,i}, \quad \mathcal{L}_p = \frac{1}{k} \sum_{t=1}^{k-1} \sum_{t=1}^{n_a} \varphi_i(g_{t+1,i}) - x_{t+1,i},$$

with total loss of $\mathcal{L} = \mathcal{L}_{ae} + \mathcal{L}_p$. The algorithm will consist of two main update steps sequentially, one to update the Koopman and the control input matrix in a distributed manner for a fixed set of GNN encoder and decoder parameters, and another to update GNN weights with a learned distributed geometric Koopman representation. Algorithm 1 shows the computational steps where the function `DistributedKoopmanMatrices(·)` presents the distributed Koopman state and input matrices, $A_D, B_D$. Thereafter the `Main(·)` function runs the update of the distributed Koopman matrices and the GNN parameters sequentially for each epoch as shown in steps 15 and 16. For simplicity of representation in the algorithm, we use the compact notations $\phi(\cdot)$ and $\varphi(\cdot)$ instead of agent-wise representation as in Figure 2.

### 3.2 Multi-Agent Network Construction via Information Transfer-based Clustering

Mapping of nodes in an NDS to nodes in an agent network is a core aspect for our proposed method. We use an information-theoretic clustering method [49] that exploits both the adjacency matrix structure as well as dynamical properties of the network for this task. For a dynamical system, the definition of information transfer [50] from a dynamical state $x_{t,i}$ to another state $x_{t,j}$ is based on the intuition that the total entropy of a dynamical state $x_{t,j}$ is equal to the sum of the entropy of $x_{t,j}$ when another state $x_{t,i}$ is not present in the dynamics and the amount of entropy transferred from $x_{t,i}$ to $x_{t,j}$. In particular, for a discrete-time dynamical system $x_{t+1} = F(x_t)$, where $x_t = [x_{t,1}^\top \quad x_{t,2}^\top]^\top$ and $F = [f_1^\top \quad f_2^\top]^\top$, the one-step information transfer from $x_{t,1}$ to $x_{t,2}$, as the system evolves from time-step $t$ to $t + 1$ is $[T_{x_{t,1} \to x_{t,2}}]_t^{t+1} = H(x_{t+1,2}|x_{t,2}) - H_{\mathcal{I}_{t,1}}(x_{t+1,2}|x_{t,2})$. Here, $H(x_{t+1,2}|x_{t,2})$ is the conditional Shannon entropy of $x_{t,2}$ for the original system and $H_{\mathcal{I}_{t,1}}(x_{t+1,2}|x_{t,2})$ is the conditional entropy of $x_{t,2}$ for the system where $x_{t,1}$ has been held frozen from from time $t$ to time

---

**Algorithm 1** Distributed Geometric Koopman Operator with Control Computation

---

1: **function** DISTRIBUTEDKOOPMANMATRICES($X_p, X_f, U, \phi$)
2:     Map the time-series data to the GNN-embedded space using $\phi$ as follows:

$$Y_p = \phi(X_p) = [\phi(x_1), \phi(x_2), \cdots, \phi(x_{k-1})], \ \ Y_f = \phi(X_f) = [\phi(x_2), \phi(x_3), \cdots, \phi(x_k)]$$

3:     **for** $\alpha = 1, 2, \ldots, n_a$ **do**
4:         Define the transformation matrices for agent $\alpha$ as:

$$T_{p,\alpha} := blkdiag(ae_1, \ldots, ae_{n_a}), \ T_{f,\alpha} := blkdiag(ee_1, \ldots, ee_{n_a}),$$
$$T_{u,\alpha} := blkdiag(eu_1, \ldots, eu_{n_a}), \ \text{where}$$
$$ae_i = (a_\alpha + e_\alpha)_i \otimes I_{m_i}, \ ee_i = (e_\alpha)_i \otimes I_{m_i}, \ eu_i = (e_\alpha)_i \otimes I_{q_i},$$

    where $\otimes$ is the Kronecker product.
5:         Compute $Y_{f,\alpha}, Y_{p,\mathcal{N}(\alpha)}, U_\alpha$ associated with agent $\alpha$ as

$$Y_{f,\alpha} = R_{f,\alpha} T_{f,\alpha} Y_f, \ Y_{p,\mathcal{N}(\alpha)} = R_{p,\alpha} T_{p,\alpha} Y_p, \ U_\alpha = R_{u,\alpha} T_{u,\alpha} U$$

6:         Solve the optimization problem: $\min_{A_{\mathcal{N}(\alpha)}, B_\alpha} \| Y_{f,\alpha} - A_{\mathcal{N}(\alpha)} Y_{p,\mathcal{N}(\alpha)} - B_\alpha U_\alpha \|_F^2$
7:         Compute $\hat{A}_\alpha = A_{\mathcal{N}(\alpha)} R_{p,\alpha}$ and $\hat{B}_\alpha = B_\alpha R_{u,\alpha}$
8:     **end for**
9:     **return**: $A_D = \begin{bmatrix} \hat{A}_1^\top & \hat{A}_2^\top & \cdots & \hat{A}_n^\top \end{bmatrix}^\top, B_D = blkdiag(\hat{B}_1, \hat{B}_2, \ldots, \hat{B}_n)$.
10: **end function**
11: **function** MAIN()
12:     Given state $(X_p, X_f)$ and input $(U)$ time-series data from a $N_a$ agent network
13:     Initialize the GNN-based encoder ($\phi$) and decoder ($\varphi$) network
14:     **for** epochs = 1,2,..., $N_{\text{epoch}}$ **do**
15:         **Koopman Update:** Run $(A_D, B_D)$ = DistributedKoopmanMatrices($X_p, X_f, U, \phi$)
16:         **GNN Update:** Compute , $\mathcal{L} = \mathcal{L}_p + \mathcal{L}_{ae}$, and backpropagate $\mathcal{L}$ to update $\phi, \varphi$ parameters.
17:     **end for**
18:     **return:** Updated $A_D, B_D, \phi,$ and, $\varphi$.
19: **end function**

---

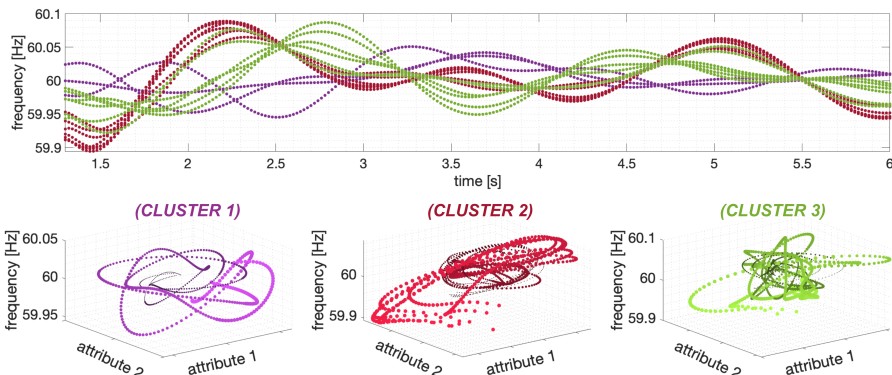

**Figure 3:** Illustration of divergent cluster dynamics from a power network. Top plot shows that the transient frequency trajectories from three different clusters behave differently. Phase-space plots in the bottom row illustrate the temporal evolution of the nodal attributes in each cluster, initial time-points are marked larger and lighter, while later time-points are marked thinner and darker.

$t + 1$. Note that the information transfer is in general asymmetric and characterize the influence of one state on any other state. Furthermore, for stable dynamical system the information transfer between the states always settle to a steady state value.

We use this information transfer measure to define an influence graph for the NDS studied in this paper. We form a directed weighted graph with the states as the nodes and introduce an edge from $x_{t,1}$ to $x_{t,2}$ iff the information transfer from $x_{t,1}$ to $x_{t,2}$ is non-zero. Moreover, the edge-weight for the edge $x_{t,1} \to x_{t,2}$ is $\exp(-|T_{x_{t,1} \to x_{t,2}}|/\beta)$ [49], where $|T_{x_{t,1} \to x_{t,2}}|$ is the absolute value of the steady-state information transfer from $x_{t,1}$ to $x_{t,2}$ (we assume stable dynamics) and $\beta > 0$ is a parameter similar to temperature in a Gibbs' distribution. Applying this to a dynamical system, a directed weighted graph is computed based on the information transfer and is clustered accordingly

to obtain a multi-agent network. Figure 3 uses a power network example to illustrate how the nodal attributes from different clusters demonstrate different transient evolution trajectories. Due to space constraints, we do not discuss the details of the data-driven computation of the information transfer from time-series data and we refer the reader to [51] for more details.

## 4 Numerical Experiments

In this section, we aim to answer the following research questions through our experiments: **(RQ1)** How does the distributed GNN-based Koopman (DKGNN) model's performance compare with other state-of-the-art approaches such as centralized GNN-based Koopman (CKGNN) [21] and graph neural network approaches for modeling multi-body interactions [52] **(RQ2)** How do various dynamical system properties such as sparsity, spatio-temporal correlation, and damping properties influence the performance boost from the distributed algorithm? **(RQ3)** What is the potential for distributed approaches for scaling to larger NDS in the future? The code for the experiments will be made online in the paper website [53][1].

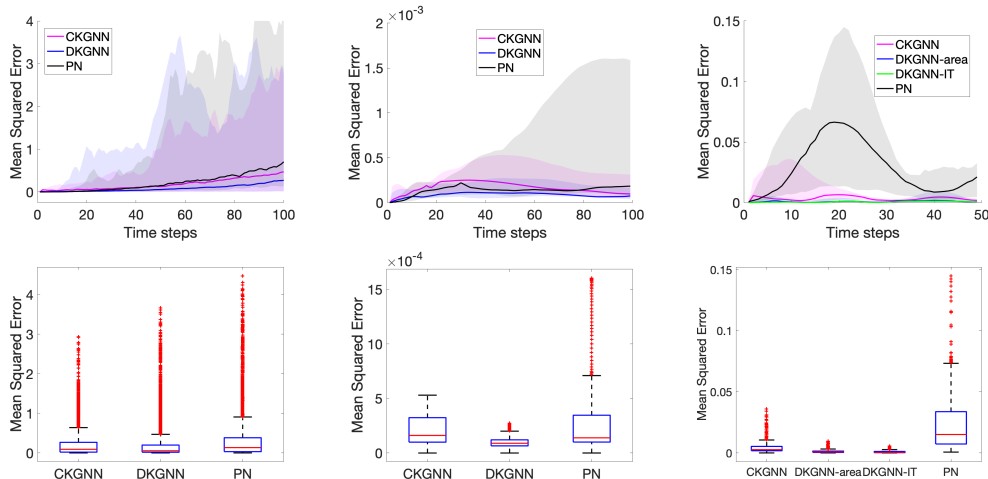

**Figure 4:** DKGNN out-performs the baselines. Columns A, B, C represent for rope, oscillator (high damping) and grid (high degree) examples, respectively. Prediction error over time-steps with darker lines representing median are shown in Row A, and MSE-based box plots are shown in Row B.

**Environments.** We perform numerical experiments on three different data-sets. The baseline rope system introduced in [7] consists of a set of six objects connected via a line graph. The second dataset is a network of oscillators which is a prototype used for modelling various real-world systems in biology [54, 55], synchronization of fireflies [56], superconducting Josephson junctions [57] etc. We consider a sparse network of oscillators consisting of 50 agents where the dynamics at each oscillator are governed by a second-order swing differential equation. The network density for this network is 0.0408. Our third dataset studies a significantly larger and complex system of immense practical importance. Power grid networks [58] are complex infrastructures that are essential for every aspect of modern life. The ability to predict transient behavior in such networks is key to the prevention of cascading failures and effective integration of renewable energy sources [59, 60]. We consider the IEEE 68-bus power grid model [61] that represents the interconnections of New England Test System and New York Power System. This is a sparse, heterogeneous network with a network density 0.0378; the network has 68-nodes, with 16-nodes representing generators and the rest being loads. The ability to accurately predict changes in voltage and frequency is key to capturing the tendency of a power grid to move towards undesired oscillatory regions. Transient stability simulations for the grid datasets are performed by the Power Systems Toolbox [62]. We provide more details on the datasets and experiments in the supplementary.

**Baselines and Implementation Details.** We baseline our method against centralized GNN-based Koopman (CKGNN) [21] and propagation network (PN), a GNN-based approach [52] using their available implementations. We evaluate all models on the trajectory prediction task where we predict

---

[1] https://sites.google.com/view/dkgnn/home

| Power Grid | CKGNN | DKGNN area-wise clustering | DKGNN IT-based clustering | PN |
|---|---|---|---|---|
| Disturbance Location | MSE | MSE | MSE | MSE |
| One hop | 0.0352 | **0.0064** | **0.0079** | 0.1123 |
| Two hops | 0.0212 | **0.0044** | **0.0049** | 0.2498 |
| Three hops | 0.029 | **0.0075** | **0.0098** | 0.0468 |
| High degree | 0.0055 | **0.0013** | **8.538E-04** | 0.0246 |
| Low degree | 0.0195 | **0.005** | **0.006** | 0.0427 |

| Oscillator | CKGNN | DKGNN | PN |
|---|---|---|---|
| Disturbance Location | MSE | MSE | MSE |
| High damping | 1.938E-04 | **1.026E-04** | 3.6E-04 |
| Low Damping | 1.404E-04 | **1.185E-04** | 5.205E-04 |
| Random | 4.86E-05 | 4.059E-04 | 4.7328 |
| **Rope** | 0.2301 | **0.2008** | 0.3091 |

**Table 1:** Prediction performance of the proposed distributed geometric Koopman approach with other baselines in terms of mean square errors (MSE) averaged over all time-steps and states for test trajectories.

the node-level time-series measurements for each of these environments that represent velocity of objects (rope), angles and frequencies of oscillators, and frequency measurements for the power grid. For PN, we slightly modify the prediction workflow from the author-provided implementation to make sure that we always feed the PN with the predicted signal values except for the initial signal input for the prediction task in consecutive time steps. Our methods are implemented on the Pytorch framework [63] and run on an NVIDIA A100 GPU. For PN baseline, we used 3 propagation steps with 32 as the batch size. The dimensions of the hidden layer of relation encoder, object encoder, and propagation effect are set as follows, for PN we use 150, 100, and 100, respectively for all the cases, for the rope system with CKGNN and DKGNN, we have used 120, 100, and 100, and for the oscillator and grid example with both CKGNN and DKGNN, these are set to be 60. The models are trained with `Adam` based stochastic gradient descent optimizer with a learning rate of $10^{-5}$. For rope, we consider $10,000$ episodes with 100 time-steps and training-testing division of $90\% - 10\%$, and batch size of 8 episodes. We consider the oscillator node state trajectories of 100 time-steps and trained with a total of 9000 time-steps, batch size of 10 trajectories, and have tested with three different testing configurations and predicting for each trajectory of 100 time-steps. Considering the power grid network, we have considered 50 time-steps to capture the initial fast transients and train the model with 1100 time steps with a batch size of 5 episodes along with testing in five different scenarios with multiple testing trajectories each with 50 time steps.

Next, RQ1 is a general discussion with respect to varying sparsity degrees, and then RQ2 is focusing more on the effect of dynamical system parameters.

**RQ1: Prediction performance analysis.** Figure 4 shows that DKGNN outperforms other baselines in trajectory prediction. Table 1 reports the mean square error (MSE) averaged over all time-steps and over all states in the test trajectory dataset. The rope system is minimally sparse with a network density of 0.33, thereby resulting in improved predictive performance for the DKGNN when compared to KGNN. The improvements are significantly pronounced for sparser and larger network models of oscillators (network density 0.0408) and power grid (network density 0.0378). The superior prediction performance over considerable trajectory time-steps (as demonstrated in Figure 4 second row) substantiates the applicability of DKGNN for sparse NDS.

**RQ2.1 Performance with respect to varying NDS properties.** The damping parameter provides us with a way to systematically study the response of an NDS to an input. Damping is a property of the NDS, and not a property of perturbations to the system. Lower damping implies that the system will take longer to converge to a steady state. We hypothesize that the introduction of input perturbations of the same magnitude at different nodes will evoke different responses depending on the connectivity structure around these nodes. For the oscillator network, we consider testing scenarios with disturbances created at high damping nodes ($> 13$ in appropriate units) and low damping nodes ($< 1$). We consider five different configurations for the power grid network. Three of the scenarios are based on the perturbations in the loads which are respectively one-hop, two-hop, and three-hop away from the generator buses, and two other load disturbance scenarios are considered at locations with high and low degrees of connectivity. The IEEE-68 bus grid network specifications also include an area-wise partitioning that is done based on eigenvalue separation and extensive application of domain-knowledge [64]. We observe DKGNN approach is able to produce better prediction performance with all of these scenarios ($81, 79, 74, 76, 74\%$ improvements with area-wise partitioning, and $77, 76, 66, 84, 69\%$ improvements with information-theoretic partitioning for the five cases listed in Table I from top to bottom with respect to the centralized approach). The second and third columns of Figure 4 correspond to the oscillator (where the disturbance is at high-damping nodes), and the power grid (where the disturbance is at high-degree buses), respectively, where both of them show the superior performance of DKGNN to the baselines.

**RQ2.2 Performance with respect to sparse NDS clustering.** This subsection reports our validation of the effectiveness of the information-theoretic clustering based agent structure discovery using the power grid network. We compare our IT-based clustering with respect to the grid's standard area-wise partitioning as discussed previously. Both the expert-driven partitioning and our clustering driven partitioning divide the grid into 5 clusters and yields a 5-agent network to use for the training. Table I and Figure 4 show that DKGNN exploits the localized dominant dynamics and yields superior predictive performance when compared to the centralized approaches such as CKGNN and PN.

**RQ3. Computational Scalability.** We compare the run time of our DKGNN with the corresponding centralized one, KGNN. Let $\tau(\phi + A_D + \varphi)$ denote the combined computation time for GNN encoder ($\phi$), distributed Koopman ($A_D$) and the GNN decoder ($\varphi$). Similarly, $\tau(\phi + A + \varphi)$ denote the computation for KGNN where the centralized Koopman ($A$) is obtained. The reduction in total run-time (%) is computed as $\frac{\tau(KGNN) - \tau(DKGNN)}{\tau(KGNN)} \times 100$. From Figure 5, it is clear there is a significant reduction in run time for the larger and sparser networks of oscillator (50 nodes with network density = 0.041), and power grid ( 68 nodes, 5 clustered areas and network density of 0.038). These examples see a considerable performance boost ($45.63\%$ for oscillator and $32\%$ for power grid) owing to capturing the dominant localized dynamic behavior. The rope which is a smaller system with 6 nodes and single excitation (at the top) shows only slightly improvement in runtime (5%), owing to high network density (0.33).

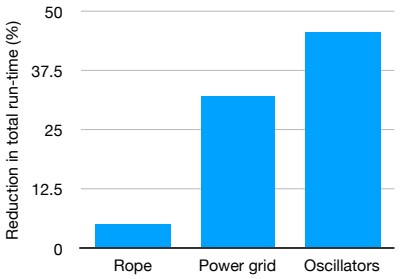

**Figure 5:** Scalability of DKGNN compared to KGNN model.

## 5 Conclusions

We present a geometric deep learning based distributed Koopman operator (DKGNN) framework that can exploit dynamical system sparsity to improve computational scalability. Our results on bounding the DKGNN performance with respect to its centralized counterpart provides a rigorous theoretical foundation. Extensive empirical studies on large NDS of oscillators and practical power grid models show the effectiveness of DKGNN design with respect to varying degree of NDS dynamical properties and sparsity patterns. Future research will look into incorporating attention capability to the distributed design, investigating the robustness aspects in presence of physical or adversarial faults, and perform control designs on the learned distributed dynamical model.

## 6 Acknowledgement

This work was supported in part by the Pacific Northwest National Laboratory (PNNL) through the Battelle Memorial Institute for the U.S. Department of Energy under Contract DE-AC05-76RL01830, and in part by the Data Model Convergence (DMC) Initiative at PNNL.

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

# 7 Appendix

**Notations.** The vectors $a_j, e_j$ respectively denote the $j^{th}$ column of the adjacency matrix, $Adj$ and a vector of standard basis in $\mathbb{R}^m$. The notation, $(a_j)_i$ denotes the $i^{th}$ entry of the column vector $a_j$. $I_m$

denote the identity matrix of size $m$. The Kronecker product is denoted by $\otimes$. The block diagonal matrix with block matrices, $M_1, M_2, \ldots, M_\ell$ is denoted by $blkdiag(M_1, M_2, \ldots, M_\ell)$.

Suppose $D = \begin{bmatrix} D_1^\top & D_2^\top & \cdots & D_\ell^\top \end{bmatrix}^\top$ where $D_1, D_2, \ldots, D_\ell$ are wide rectangular matrices. Then from the definition of the Frobenius norm, we have,

$$\| D \|_F^2 = \sum_{i=1}^{\ell} \| D_i \|_F^2 . \tag{S1}$$

If $D_1$ and $D_2$ are two matrices, then

$$\| D_1 - D_2 \|^2 = \| D_1 \|^2 + \| D_2 \|^2 - 2\mathrm{trace}(D_1^\top D_2) \tag{S2}$$

where $\mathrm{trace}(D_1^\top D_2)$ denote the Frobenius inner product of the matrices $D_1$ and $D_2$. For any matrix $D$, the Moore-Penrose inverse is denoted by $D^\dagger$.

### 7.1 Proof of Proposition 3

*Proof.* Consider the the centralized learning problem described in Eq. M3 (where the notation 'M' indicates the equation is from the main manuscript). This problem is now rewritten with respect to each agent as follows.

$$
\| Y_f - AY_p - BU \|_F^2 = \left\| \begin{bmatrix} Y_{f,1} - A_1 Y_p - B_1 U_1 \\ Y_{f,2} - A_2 Y_p - B_2 U_2 \\ \vdots \\ Y_{f,n_a} - A_{n_a} Y_p - B_{n_a} U_{n_a} \end{bmatrix} \right\|_F^2 = \sum_{\alpha=1}^{n_a} \| Y_{f,\alpha} - A_\alpha Y_p - B_\alpha U_\alpha \|_F^2 \quad \text{(from Eq. S1)}
$$

$$
= \sum_{\alpha=1}^{n_a} \| Y_{f,\alpha} - A_{\mathcal{N}(\alpha)} Y_{p,\mathcal{N}(\alpha)} - B_\alpha U_\alpha - A_{\overline{\mathcal{N}(\alpha)}} Y_{p,\overline{\mathcal{N}(\alpha)}} \|_F^2 \quad \text{(from Remark 2)}
$$

$$
= \sum_{\alpha=1}^{n_a} \| Y_{f,\alpha} - A_{\mathcal{N}(\alpha)} Y_{p,\mathcal{N}(\alpha)} - B_\alpha U_\alpha \|_F^2 + \| A_{\overline{\mathcal{N}(\alpha)}} Y_{p,\overline{\mathcal{N}(\alpha)}} \|_F^2
$$

$$
- 2\,\mathrm{trace}\left( (Y_{f,\alpha} - A_{\mathcal{N}(\alpha)} Y_{p,\mathcal{N}(\alpha)} - B_\alpha U_\alpha)^\top A_{\overline{\mathcal{N}(\alpha)}} Y_{p,\overline{\mathcal{N}(\alpha)}} \right) \text{(from Eq. S2)}
$$

$$
= \sum_{\alpha=1}^{n_a} \| Y_{f,\alpha} - A_{\mathcal{N}(\alpha)} Y_{p,\mathcal{N}(\alpha)} - B_\alpha U_\alpha \|_F^2
$$

where the last step follows by noticing that $A_{\overline{\mathcal{N}(\alpha)}} = 0$ since the agent $\alpha$ is not connected to the agents in $\overline{\mathcal{N}(\alpha)}$. In the above steps, the computation of $Y_{f,\alpha}$ and $Y_{p,\mathcal{N}(\alpha)}$ involves computing the transformation matrices $T_{f,\alpha}, R_{f,\alpha}, T_{p,\alpha}, R_{p,\alpha}$ which are computed under the knowledge of the network topology.

Finally, we obtain,

$$
\min_{A,B} \| Y_f - AY_p - BU \|_F^2 = \min_{\substack{A_{\mathcal{N}(\alpha)}, B_\alpha \\ \alpha \in \{1,2,\ldots,n_a\}}} \sum_{\alpha=1}^{n_a} \| Y_{f,\alpha} - A_{\mathcal{N}(\alpha)} Y_{p,\mathcal{N}(\alpha)} - B_\alpha U_\alpha \|_F^2
$$

$$
= \sum_{\alpha=1}^{n_a} \min_{A_{\mathcal{N}(\alpha)}, B_\alpha} \| Y_{f,\alpha} - A_{\mathcal{N}(\alpha)} Y_{p,\mathcal{N}(\alpha)} - B_\alpha U_\alpha \|_F^2
$$

For every $\alpha \in \{1, 2, \ldots, n_a\}$, $A_{\mathcal{N}(\alpha)}, B_\alpha$ can now be obtained analytically as $\begin{bmatrix} A_{\mathcal{N}(\alpha)} & B_\alpha \end{bmatrix} = Y_{f,\alpha} \begin{bmatrix} Y_{p,\mathcal{N}(\alpha)} & U_\alpha \end{bmatrix}^\dagger$ and the transition mapping corresponding to the agent $\alpha$ is given by

$$\hat{A}_\alpha = A_{\mathcal{N}(\alpha)} R_{p,\alpha}, \quad \text{for } \alpha \in \{1, 2, \ldots, n_a\}.$$

Finally the distributed Koopman is given by $A_D = \begin{bmatrix} \hat{A}_1^\top & \hat{A}_2^\top & \cdots & \hat{A}_{n_a}^\top \end{bmatrix}^\top$, input matrix is $B_D = blkdiag(B_1, B_2, \ldots, B_{n_a})$. Hence the proof. $\qquad\square$

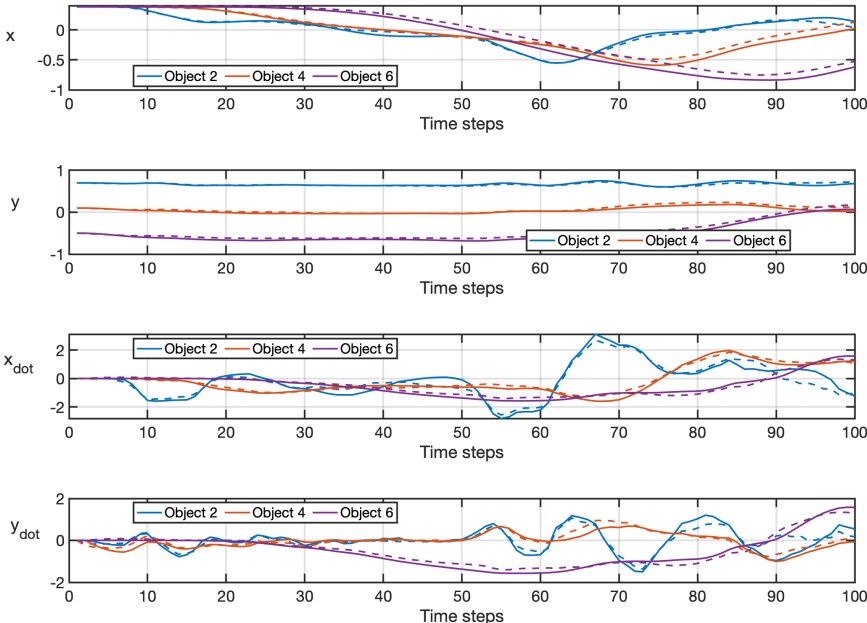

**Figure 6:** Rope prediction performance under random control inputs (the solid lines indicate the actual state trajectories and the dotted line shows the corresponding predictions.)

### 7.2 Proof of Corollary 4

*Proof.* The proof follows by noticing that in a fully connected network, for any agent $\alpha$, the corresponding non-neighbors set, $\overline{N(\alpha)}$ is empty. $\qquad\square$

### 7.3 More Details on the Numerical Studies

The network topologies of the three example systems are given in Figure 7. We follow the physics simulation engine provided by [21] for generating the data for the rope example, and follow the baseline node and edge attributes for the objects and connecting edges.

For the oscillator network, each of the individual node dynamics follows a second order differential equation. The overall dynamics is represented as:

$$\begin{bmatrix} \dot{\theta} \\ \ddot{\theta} \end{bmatrix} = \begin{bmatrix} 0_{n_v} & I_{n_v} \\ -\beta M^{-1}\mathcal{L} & M^{-1}D \end{bmatrix} \begin{bmatrix} \theta \\ \dot{\theta} \end{bmatrix}, \tag{6}$$

where $\theta \in \mathbb{R}^{n_v}, \dot{\theta} \in \mathbb{R}^{n_v}$ are the angles and frequencies of the oscillator. The diagonal matrices $M$ and $D$ contain inertia and damping of the nodes. The coupling of the nodes are captured by the Laplacian $\mathcal{L}$ with their strengths represented by $\beta$. $0_{n_v}$ and $I_{n_v}$ denote the zero and identity matrices of size $n_v$. For the oscillators we have created one-hot vector for node attributes. We have divided the nodes into low inertia ($< 3$ in appropriate units), medium inertia ($> 3$, but $< 8$), and considerably high inertia ($> 8$), thereby creating $3-$dimensional node attribute vectors. The edge attributes are also one-hot vectors with 6 different types. Based

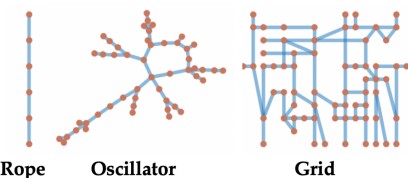

**Figure 7:** Network topologies

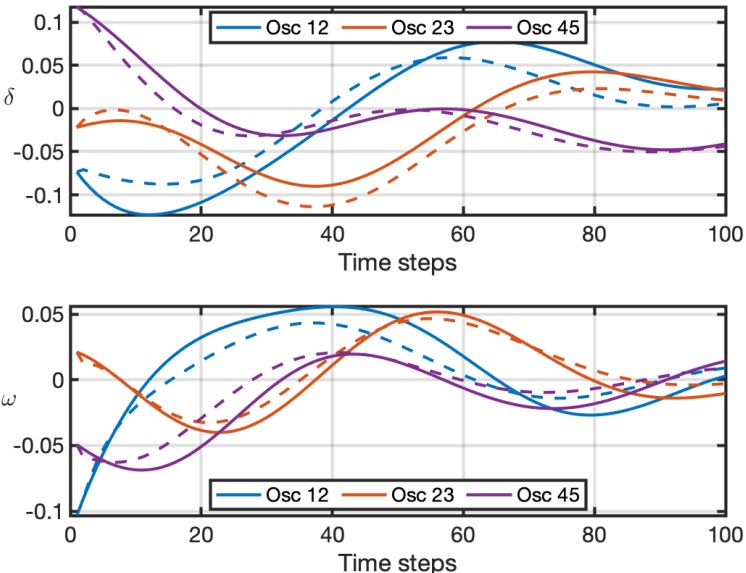

**Figure 8:** Oscillator prediction performance with random perturbations where the model is distributed according to the adjacency matrix of the underlying network (the solid lines indicate the actual state trajectories and the dotted line shows the corresponding predictions.)

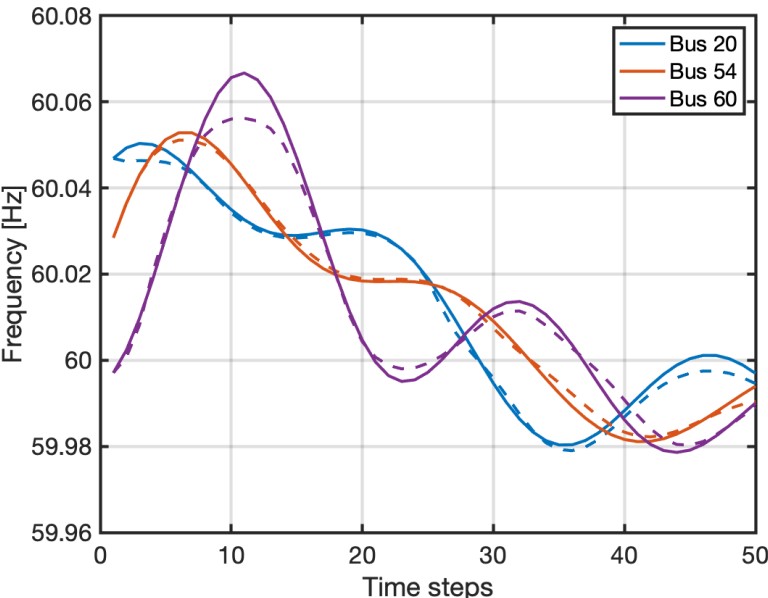

**Figure 9:** Powergrid prediction performance with information theoretic clustering and two-hop load perturbations (the solid lines indicate the actual state trajectories and the dotted line shows the corresponding predictions.)

on inertia, these types are: low-low, high-high, medium-medium, and un-directed low-medium, medium-high, and low-high.

The IEEE benchmark 68-bus powergrid model is simulated with detailed dynamics following the power system toolbox [62]. Our prediction models are learned for fast transient frequencies. The node attributes are designed similarly as the oscillator studies, however, here the nodes are physical

powergrid buses which are classified as generator buses, load buses, and buses without any loads or generators (let us call it none), thereby creating $3-$dimensional one-hot vectors. The edges connecting buses represent powergrid transmission lines, thereby, edge attributes are characterized as generator-load, load-none, and generator-none connections, resulting in a $3-$dimensional one-hot vectors.

The network density of the systems are used to characterize the sparsity which is defined as the ratio between the number of edges to the maximum number of possible edges.

We present few examples of the prediction performance of the DKGNN model for predicting the dynamic system behaviours over time steps. Figure 6 shows the the positions and velocities of the rope objects $2, 4$ and $6$ for the predicted performance with respect to the actual physics simulations. Figure 8 shows an example of prediction performance for the network of oscillators at nodes $12, 23,$ and $45$. Prediction performance for the powergrid example is shown in Figure 9 with information-theoretic clustering used for the DKGNN for buses $20, 54,$ and $60$. These figures show satisfactory performance for the distributed geometric Koopman models for all the networked dynamic system examples.

