# OpenReview forum: "Learning Distributed Geometric Koopman Operator for Sparse Networked Dynamical Systems"
_logconference.io/LOG/2022/Conference — LoG 2022 Poster_

### Official Review · Reviewer_f7Mp · 2022-10-17

**Overall Score:** 6
**Confidence:** 4

**Review:**

### **Summary**:
This paper presents a learnable simulation algorithm for Networked Dynamical Systems (NDS) based on Koopman operators. In particular, Koopman operator theory asserts that a non-linear dynamical system can be reparametrised via a set of “observables”, the dynamics of which are linear (and are governed by the infinite-dimensional Koopman operator). Following common practices in the Koopman operator learning literature, the authors learn a finite-dimensional Koopman operator, by lifting the system state in a vector of observables and instantiate the Koopman operator as a matrix that is multiplied with this vector.

The main innovation in this paper, is what the authors call “distributed Koopman”, i.e., a specific parametrisation of the Koopman operator as a block diagonal matrix, based on the sparsity of the system. The authors first perform a clustering algorithm on the network and then bestow the resulting sparsity pattern on the Koopman matrix. They show that this allows solving a Koopman matrix prediction problem independently for each cluster and thus significantly improve scalability. This is verified experimentally across several NDS benchmarks (rope, oscillators, power-grids), while an improvement in simulation accuracy is also shown, especially in the presence of high sparsity.


### **Main Review**:

**Strengths**:

- **Relevance**: The authors identify an important characteristic of many Network DS, that of sparsity, which plays an important role in the resulting dynamics, but has so far been mostly overlooked in the literature.
- **Impact**: Introducing sparsity in Koopman operator learning is a natural and simple idea that is illustrated to work well in practice, since it significantly simplifies (and accelerates) the optimisation problem, while simultaneously introducing a stronger inductive bias that improves generalisation. At the same time, it makes Koopman more scalable and therefore widens the applicability of the method
- **Presentation/Execution**: The idea is adequately presented with several accompanying explanatory figures, while the experimental section is well-executed and the results show the benefits of the approach (average and long-term simulation accuracy, scalability)


**Weaknesses**:

- **Novelty**: Despite the simplicity of the method, I am a bit concerned about its novelty. In particular, Koopman learning has been explored in several papers, including applications to Networked DS (e.g., Li et al., ICLR'20, where GNNs are used to model the lifting to the observables). Moreover, the authors use a predefined algorithm for graph clustering from Sinha ECC’22 . Therefore, it seems to me that the innovation here is the use of the sparsity patterns in the Koopman matrix. Of course, I don't consider this as grounds for rejection, but I do think that the topic of Koopman operators for Networked DS has potential for a deeper study (see below for suggestions), which would strengthen the novelty and impact of the manuscript.
- **Experimental comparisons**:
   - Comparison against Li et al., ICLR’20: The authors mention this method as Centralised Koopman GNN (CKGNN) in their experimental section, but in this paper, the authors use object-centric embeddings, while CKGNN seems to be using system-centric embeddings. Isn’t this more closely-related to the approach of Yeung et al., ACC’19? Could the authors clarify this? I believe a comparison with the exact formulation of Li et al. should be included as well.
   - Including more baselines and ablating the different lifting functions learned would be beneficial to strengthen the claims of this paper. For example, it might be useful to also compare against handcrafted Koopman basis functions (as in Li et al., and Yeung et al.) and against other neural architectures (e.g., MLP) as lifting functions, in order to better understand the benefits of a graph-based representation (as in Yeung et al., Lusch et al., Nature Comm.’18 and others). Lusch et al. additionally parameterise the eigenvalues of the Koopman matrix as a function of the observables, which might of possible interest for a more complete comparison.
  - I would also encourage the authors to better motivate the necessity of using Koopman operator instead of simulating directly in the original state space. From the experimental section, it seems that PN perform consistently worse than Koopman-based methods. Could the authors comment on this? It might also make sense to compare against more sophisticated architectures that operate in the original state space (e.g., more expressive GNN architectures, or GNN architectures that also operate on the clusters, hierarchical formulations as in Mrowca et al., NeurIPS’18). It would be ok not to include additional experimental results, but at least a discussion would help.

**Questions and additional Suggestions**:

- **Clustering**: It seems that a core component of the method is the clustering algorithm used in order to infer the block diagonal adjacency matrix (the multi-agent representation as referred to by the authors).
   - Why did the authors use this particular algorithm to compute the edge weights and how sensitive are the results when choosing other graph constructions? The authors also mention that they construct the graph based on the “steady-state information transfer”. What do the authors mean by this and how is this computed? Would it be possible to construct the graph dynamically?
   - How did the authors perform the clustering of the graph after computing the edge weights? Is it k-means as in Sinha, ECC'22? How sensitive is the method to different clustering algorithms (e.g., other alternatives are Stochastic Block Models, Louvain, or learned clustering, e.g., Wilder et al., NeurIPS'19, Locatello et al., NeurIPS'20, Bouritsas et al., NeurIPS'21)?
   - Would it be possible to also learn a clustering algorithm together with the GNN lifting/Koopman matrices (see above for ideas to do this)?

- **Implementation**:
   - Do the authors use the analytical solution for the Koopman operators during training? Aren’t there practical issues with this, e.g., high memory footprint, slow computation of the inverse, numerical instabilities etc.?
   - Would it be possible to use an invertible network for the lifting?
   - When unrolling trajectories, is it necessary to project each state back to the space of observables, or is it possible to unrol the dynamics in the lifted space and only decode? In Lusch et al., Nature Comm.'18, the authors use an additional loss in the lifted space, presumably for this functionality. Could the authors comment on that?


- **Evaluation**:
   - In Table 1, do the authors evaluate on unrolled trajectories or on one-step prediction?
    - Clarification on Fig 4 (top row): What do the shaded regions represent (standard deviation, range?) It seems to me that distributed Koopman has a larger variability in the rope experiment. Could the authors comment on that?

**Minor**:
- **Clarity**: I think the clarity of the text can be improved in certain parts, to improve readability. For example, notation could be simplified/off-loaded (e.g., Algorithm 1 is notation-heavy and becomes hard to follow). Moreover, section 3.2. uses notions from Sinha ECC’22 that are not appropriately explained (e.g., how are the entropy terms estimated, what is the steady-state information transfer).
- **Theoretical results**: The theoretical results (although useful) can be directly derived from the block-diagonal formulation of the Koopman matrix, so it might be more appropriate to avoid stating them as a theorem, since this creates an anticipation for something more substantial.

### **Overall**:
The paper presentes an interesting and well-executed method for simulating Networked Dynamical Systems, taking into account sparsity, a central characteristic of many NDS. The approach is technically sound and the results corroborate the authors' claims. Although, as mentioned above, I believe that there are several routes to deepen this study, this topic deserves more attention from the community, and I will therefore vote for acceptance.

---

### Official Review · Reviewer_FoEx · 2022-10-22

**Overall Score:** 5
**Confidence:** 3

**Review:**

### Summary

The paper introduces a method for learning the dynamics of a networked dynamical system. The nodes of the graph are clustered into agents and the Koopman operator is learned for each in a distributed fashion. This exploits the topology and sparsity of the graph in order to scale efficiently to larger graphs. Experiments are conducted on three different types of systems, where the proposed method shows improved prediction accuracy and runtime.

### Strengths

* The paper develops a solid approach to the challenging and important problem of learning networked dynamics. In particular, the method is the first to leverage graph coarse-graining and sparsity, which is very important when considering large networked dynamical systems.

* The information-thereotic clustering is a clever and effective way of segmenting the graph.

* The experiments are thorough, with a good diversity of network types and initial conditions tested.

### Content concerns

* The key assumption of the paper is that state update dynamics on the "cluster graph" are dependent on immediate neighbors only (169-171). However, it is not clear to me why this would be true. Perhaps it is a useful heuristic that works well in practice, but the paper makes it a key premise of the theory. This calls into question the main thereotical result since $A_\bar{\mathcal{N}(\alpha)} \neq 0$ cannot be assumed.  For example, for the coupled oscillators, this is not true for any finite time step.

* Instead of asserting the sparsity and/or taking it for granted, it would seem more correct to _postulate_ that that the Koopman operator becomes sparse as cluster size increases, and then explore the tradeoff between (1) the accuracy of the approximation and (2) the size of the state spaces, to find the best resolution scale.

* Is the main message-passing on the original graph or the agent/cluster graph? Either way, it seems that the transformed state of an agent $g_t$ is a function of not only the original state of the agent, but the neighbors of its constituent nodes that are in other clusters, and (with enough message passing layers) the entire NDS. This again seems to counter the assumption that the dynamics are dependent on immediate neighbors only, since the even the state itself may not be.

* Does the Koopman operator theory hold under these local decompositions? That is, given locally governed dynamics, is it true that the dynamics of local observables in function space are also locally governed?


### Presentation concerns

* Did you explore different settings / thresholds for the clustering? Please provide further statistics on the size and number of final clusters. How did you approximate the conditional entropy?

* Why is the control theory aspect necessary and what are the control inputs? The experiments appear to involve only predicting the evolution from a perturbed state. Also, why are you calling clusters agents? There is no action space as far as I can tell.

* The discussion of the results is poorly organized.  RQ1 and RQ2 are not meaningfully different, and the "spatio-temporal correlation" (255) aspect of RQ2 is never discussed. The "damping parameter" (304) is introduced in a way that suggests the term encompasses power grid pertubrations, but it does not. Area-wise paritioning is mentioned (315) before it is defined (322).

### Recommendation

The paper addresses an important problem, and I like the idea and empirical results. However, since the validity of the thereotical claims remains uncertain, I cannot recommend acceptance. If my concerns are mistaken or the authors address them thoroughly, or the authors rephrase their claims, I am happy to recommend acceptance.

---

### Official Review · Reviewer_kDpK · 2022-10-22

**Overall Score:** 5
**Confidence:** 2

**Review:**

This paper proposes to model the dynamical system with graphs. Specifically, the objects are coarsened into clusters (called agents in this paper), and then use graph neural networks to encode neighboring agents to predict the dynamics. This framework can reduce the computational complexity such that enables scalability. Also, it allows each agent to preserve their own dynamics, which can be diverse and personalized.

\

### Pros
- Modeling dynamical systems with networks (graphs) is a natural and reasonable idea. Graph neural networks are also fair tools to capture the interaction between objects in such systems.
- Incorporating the traditional Koopman Operator with the current deep learning models is an interesting attempt.
- The writing is clear and easy to follow.

### Suggestions and questions
- The claim of "distributional" may not be rigorous. In a distributed system, if I'm correct, an agent (client) can not directly access the data from others. However, the "Distributed Geometric Koopman Model" (fig. 2) in this paper violates this principle - it seems the GNN is a whole model for all agents. Using GNN is natural, but it may not be appropriate to claim this work as "distributed".
- One major benefit of the proposed method is scalability. This is pretty important in real-world applications, especially graph settings. However, I would expect a more theoretical complexity analysis. For example, what are the time/space complexities? Are the complexities better than baselines?
- Is the proposed clustering method better than other existing clustering approaches? The authors can add one experiment to compare their design with other node clustering methods.
- How to define "divergent" cluster dynamics? Is there a criterion that can differentiate "divergent" dynamical systems from their "non-divergent" counterparts?
- The NDS in this work is still in discrete times. The authors may consider discussing the continuous time such as [1] as well. For example, what's the potential of Koopman in continuous time.


[1] Huang, Zijie, Yizhou Sun, and Wei Wang. "Learning continuous system dynamics from irregularly-sampled partial observations." Advances in Neural Information Processing Systems 33 (2020): 16177-16187.

---

### Meta-Review · Area_Chair_absu · 2022-11-20

**Confidence:** 3
**Recommendation:** Reject

**Meta Review:**

This paper proposes to scale Koopman operator for sparse NDS in a distributed setting, where the original graph is coarsened via clustering. The reviews are mixed. Strengths include:
S1: This paper addresses an important problem for NDS.
S2: The proposed solution is reasonable.
Weaknesses include:
W1: The novelty is mainly on the scalability side, as both Koopman learning and the clustering algorithm are studied by existing papers.
W2: Better motivation of Koopman operator is needed, especially to compare with other ways to handle NDS.
W3: The experimental results need better justification.
Some of the concerns are partially addressed during the rebuttal, but no reviewer changed their score. Weighing the pros and cons, the paper needs further enhancement for publication.

---

### Decision · Program_Chairs · 2022-11-22

**Decision:**

Accept (Poster)

**Comment:**

We discussed this among the PCs and found that there are no substantial issues barring the paper from being presented at the conference. We strongly encourage the authors to incorporate reviewer and AC comments in their camera-ready revision; we believe that this will fundamentally strengthen the paper and extend its visibility.